# ZARA: Improving Few-Shot Self-Rationalization for Small Language Models

**Wei-Lin Chen**[1]  **An-Zi Yen**[2]  **Cheng-Kuang Wu**[1]  **Hen-Hsen Huang**[3]  **Hsin-Hsi Chen**[1]

[1]National Taiwan University, Taiwan
[2]National Yang Ming Chiao Tung University, Taiwan
[3]Academia Sinica, Taiwan
wlchen@nlg.csie.ntu.edu.tw
hhchen@ntu.edu.tw

## Abstract

Language models (LMs) that jointly generate end-task answers as well as free-text rationales are known as self-rationalization models. Recent works demonstrate great performance gain for self-rationalization by few-shot prompting LMs with rationale-augmented exemplars. However, the ability to benefit from explanations only emerges with large-scale LMs, which have poor accessibility. In this work, we explore the less-studied setting of leveraging explanations for small LMs to improve few-shot self-rationalization. We first revisit the relationship between rationales and answers. Inspired by the implicit mental process of how human beings assess explanations, we present a novel approach, Zero-shot Augmentation of Rationale-Answer pairs (ZARA), to automatically construct pseudo-parallel data for self-training by reducing the problem of plausibility judgement to natural language inference. Experimental results show ZARA achieves SOTA performance on the FEB benchmark, for both the task accuracy and the explanation metric. In addition, we conduct human and quantitative evaluation validating ZARA's ability to automatically identify plausible and accurate rationale-answer pairs.[1]

## 1 Introduction

Driven by the concerns of whether the decisions made by the artificial intelligence models are trustworthy, providing free-text, natural language explanations (NLEs) has drawn substantial attention in the research community (Camburu et al., 2018; Li et al., 2018; Rajani et al., 2019; Aggarwal et al., 2021; Chen et al., 2022). Comparing with popular explanation techniques within the input scope, e.g., attributing feature importance scores to tokens (Li et al., 2016; Godin et al., 2018) or extracting fragments of text highlights (Lei et al., 2016; Jain

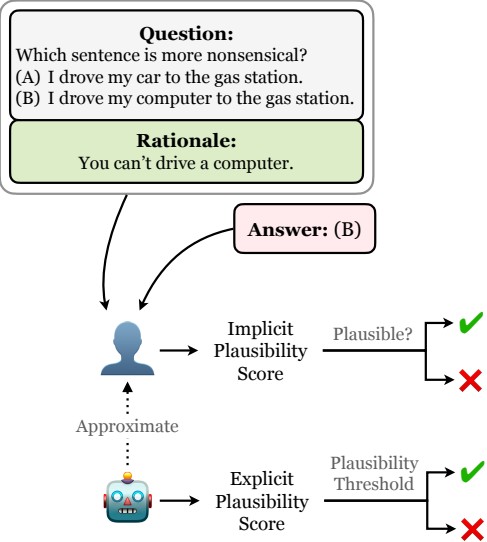

Figure 1: The role of our plausibility agent. As the main component of ZARA, the agent, i.e, the approximator, imitates how humans assess the plausibility of explanations, in an *explicit* fashion.

et al., 2020), free-text explanation[2] is more expressive, inherently apt for human comprehension and brings richer information in addition to input context (Camburu et al., 2018; Wiegreffe et al., 2021). Yet, the construction of NLE datasets is expensive and challenging due to quality control issues such as inconsistency and under-specification (Wiegreffe and Marasovic, 2021). The development of interpretable NLP systems which can provide NLEs in few-shot is necessitated.

Recent works (Wei et al., 2022; Wang et al., 2022b; Lampinen et al., 2022) achieve few-shot self-rationalization, i.e., jointly generating free-text explanations and end-task labels, by extending the usage of NLEs to compose *chain-of-thought* (CoT) input-rationale-output demonstrations for prompt-based learning. Comparing with standard

---

[1]https://github.com/ntunlplab/ZARA

[2]We use the term "free-text explanation" and "natural language explanation" interchangeably; and the term "explnantion" can also refer to the rationale generated by the model.

prompting (i.e., without rationales), prompting with rationale-augmented exemplars triggers LM's complex reasoning ability, significantly boosting the end-task performance. However, the main drawback is that only excessively large LMs (generally 100B-plus) demonstrate this ability to leverage explanations, which sharply emerges when scaling model size sufficiently (Wei et al., 2022; Lampinen et al., 2022).

In this work, we explore the less-studied setting of improving few-shot self-rationalization only relying on affordable, small LMs (200M∼2.7B). We adopt *self-training* (Scudder, 1965), a simple yet effective methodology that is not practical for large LMs in most real-world scenarios. We first investigate the relationship between the generated explanations and end-task predictions, and find plausible explanations are usually paired with correct label predictions. Namely, plausibility is a strong indicator for answer correctness. Motivated by this finding, we propose **Z**ero-shot **A**ugmentation of **R**ational-**A**nswer pairs (ZARA) for self-training.

Specifically, we reduce the problem of assessing rationale plausibility to the task of natural language inference (NLI), and propose a zero-shot *plausibility approximator* towards automatic assessment of the generated rationales, without requiring any ground-truth labels or golden explanations. The approximator can be viewed as an agent for plausibility judgement. As illustrated in Figure 1, to determine the plausibility of the rationale, humans implicitly ask themselves whether they can draw conclusions to the predicted answer by understanding the task, the input question, and the supported rationale with their logic and reasoning. To approximate such process explicitly, the approximator leverages the ability of textual entailment to yield a probability score indicating the explanation plausibility. Connecting to the self-training paradigm, we first train a self-rationalization model by few-shot prompt-based learning with natural language prompts, and leverage the approximator to collect pseudo-parallel data, i.e, unlabeled inputs paired with high-confident rationale-answer pairs, for creating an augmented training set which is then used to learn an improved self-rationalization model.

With various small-size LMs, experiments show our approach notably improves the FEB benchmark[3] (Marasovic et al., 2022)—a recently pro-

posed standardized few-shot self-rationalization benchmark—with 3.4%∼5.1% and 3.0%∼5.8% for task accuracy and the associated explanation metric, respectively. Additionally, we validate the approximator's ability with both human and quantitative evaluations. The results suggest our approximator can effectively select plausible explanations that lead to higher accuracy for end-task predictions. In summary, our main contributions are three-fold:

1. We show how to leverage explanations for small LMs by an in-depth analysis of the relationship between rationales and task labels.

2. We propose ZARA, a novel approach for small LMs to improve self-rationalization with self-training.

3. Our NLI-based approximator sheds light on the potential of automatic evaluation for explanation plausibility and post-hoc verification for label accuracy.

## 2 Background and Motivation

Given a trained self-rationalization model $f_\theta(\cdot)$ and an input sequence $x$, we denote a prediction $f_\theta(x) = (\hat{r}, \hat{a})$, where $\hat{r}$ is the generated free-text rationale and $\hat{a}$ is the predicted answer, typically a classification label. Note that $\hat{r}$ and $\hat{a}$ are parsed from the output sequence of $f_\theta(x)$. Evaluation of a self-rationalization model requires assessing both $\hat{a}$ for the end-task performance and $\hat{r}$ for the quality of the explanation. With the lack of an ideal and unified automatic metric, the current gold standard for determining the quality of $\hat{r}$ is to conduct a human evaluation to check its *plausibility* (Marasović et al., 2020; Kayser et al., 2021; Wiegreffe et al., 2022; Marasovic et al., 2022). An ideal $\hat{r}$ is considered to be plausible if it is able to *justify* $\hat{a}$, that is, providing a logical and reasonable explanation supporting the model's prediction.

However, if $\hat{r}$ is deemed plausible by humans, it does not mean $\hat{a}$ is correct. As the example in Table 1, commonsense would know "*bed*" is likely the answer, yet the generated explanation for the corresponding prediction "*couch*" is still plausible. Plausibility illustrates the degree of convincement towards the model's prediction, regardless of whether the model is actually making an accurate prediction or not (Jacovi and Goldberg, 2021).[4]

---

[3]https://github.com/allenai/feb (Licensed under the Apache License 2.0.)

[4]For a confounder-free setting, prior works (Kayser et al.,

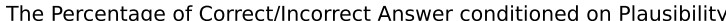

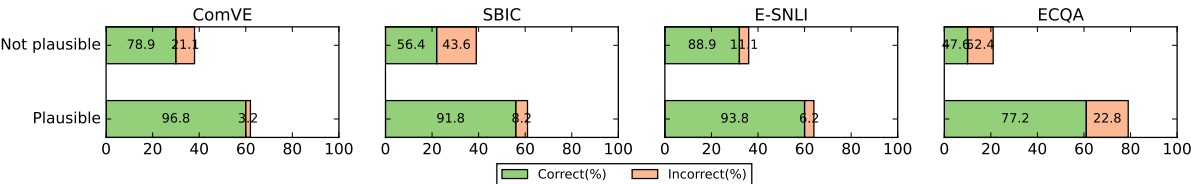

Figure 2: The percentage of correct/incorrect answers when explanations are plausible and not plausible. The y-axis denotes the percentage of explanations deemed plausible/not plausible by humans.

| Instance |
| --- |
| ***Question:*** Danny is a human. He needed to catch a flight, but stayed too long in an object made for humans to relax in. Where might he have been? 
 ***Choices: a.*** bathroom ***b.*** couch ***c.*** airport ***d.*** homes ***e.*** bed |
| ***Prediction*** ($\hat{a}$): couch 
 ***Label*** ($a$): bed 
 ***Explanation:*** Couch is an object made for humans to relax in. Danny might have been in couch. |

Table 1: A plausible but inaccurate prediction of an instance from ECQA dataset output by our model.

Naturally, generating plausible explanations that can justify the wrong answers should be much harder comparing to justifying the correct answers. Since such $\hat{r}$ usually demonstrates a slight pivot from commonsense yet still introduces a sound reason to support the inaccurate $\hat{a}$. We hypothesize this—plausible explanation towards inaccurate end-task prediction—is not the circumstance in most cases of $(\hat{a}, \hat{r})$. In other words, if $\hat{r}$ is considered to be plausible, it is likely that $\hat{a}$ is a correct prediction. Hence, the first research question arises: ***RQ1***: "To what extent do plausible explanations imply correct label predictions?" And if we could verify ***RQ1***, the follow-up question would be ***RQ2***: "Is it possible to automatically identify plausible $\hat{r}$ and utilize $(\hat{r}, \hat{a})$ for further model improvement?"

In the following of our work, we answer ***RQ1*** by inspecting the interrelationship between the plausibility of $\hat{r}$ and the correctness of $\hat{a}$ (Section 4), where we show evidence supporting the linkage to ***RQ2***. Ergo, we propose ZARA coupled with self-training to accomplish ***RQ2*** (Section 5), improving few-shot self-rationalization models.

## 3  Datasets and Tasks

We adopt FEB (Marasovic et al., 2022), a newly proposed few-shot self-rationalization benchmark,

---

2021; Marasovic et al., 2022) only evaluates $\hat{r}$ with $\hat{a} = a$, i.e, explanation for the correctly predicted answer. This may overestimate the quality of explanations (Wiegreffe et al., 2022).

as the dataset for experiments throughout this work. FEB consists of four sub-tasks from existing English-language explainable datasets with free-text explanations: (1) **Nonsensical sentence selection** (COMVE; Wang et al., 2019). Given two sentences, select the sentence that is less likely to make sense. (2) **Offensiveness classification** (SBIC; Sap et al., 2020). Classify a given post as offensive or not. (3) **Natural language inference** (E-SNLI; Camburu et al., 2018). Classify the relationship between two sequences as entailment, neutral, or contradiction. (4) **Multiple-choice commonsense QA** (ECQA; Aggarwal et al., 2021). Given a question, select the correct answer from five choices.

The goal for each sub-task is the same, namely, to predict a label for the underling classification task and generate a free-text explanation supporting the model's decision. Each sub-task has 60 episodes, and each episode is a train-test split with 48 training examples and 350 evaluation examples. This design of no extra validation data encompasses the FLEX principles (Bragg et al., 2021) for performing robust few-shot NLP evaluation to avoid per-episode hyper-parameter tuning, which could inflate the evaluation results considerably as shown in previous work (Gao et al., 2021). Hence, a single set of hyper-parameter is used across all episodes.

## 4  Correlation between Plausibility and Correctness

As described in Section 2, following we attempt to answer ***RQ1*** by measuring the correlation between the plausibility of $\hat{r}$ and the correctness of $\hat{a}$. We conduct human studies on results from a self-rationalization model (without self-training) using the FEB dataset. We adopt prompt-based fine-tuning with natural language prompt on a sequence-to-sequence language model to perform few-shot self-rationalization.

For each episode of the sub-task, we train a self-

| Mapping and Example | |
|---|---|
| **CoMVE** | **SBIC** |

**Mapping and Example**

<table>
<tr><td colspan="2">Mapping and Example</td></tr>
</table>

**ComVE**

**Example**
*choice1*: i drove my car to the gas station.
*choice2*: i drove my computer to the gas station.
*Answer*: choice2
*Rationale*: you can't drive a computer.

**Mapping**
*Premise*: [Rationale]
*Hypothesis*: [Answer's sentence]
*NLI Class*: Contradiction

**Mapped example**
*Premise*: you can't drive a computer.
*Hypothesis*: i drove my computer to the gas station.

**SBIC**

**Example**
*Post*: just when i thought women couldn't get any stupider.
*Answer*: offensive
*Rationale*: this post implies that women are stupid.

**Mapping**
*Premise*: The post: [Post]
*Hypothesis*: The post is [Answer] because [Rationale]
*NLI Class*: Entailment

**Mapped example**
*Premise*: The post just when i thought women couldn't get any stupider.
*Hypothesis*: The post is offensive because this post implies that women are stupid.

**E-SNLI**

**Example**
*Premise*: a woman in a black mesh skirt plays acoustic guitar.
*Hypothesis*: a woman is wearing black.
*Answer*: Entailment
*Rationale*: The woman is wearing a black mesh, she is wearing black.

**Mapping**
*Premise*: [Premise] [Rationale].
*Hypothesis*: [Hypothesis]
*NLI Class*: [Answer]

**Mapped example**
*Premise*: A woman in a black mesh skirt plays acoustic guitar. The woman is wearing a black mesh, she is wearing black.
*Hypothesis*: A woman is wearing black.

**ECQA**

**Example**
*Question*: what is a place that has a bench nestled in trees?
*Choices*:(a) state park (b) bus stop (c) bus depot (d) statue (e) train station
*Answer*: (a)
*Rationale*: state park is a protected public garden. public gardens generally have benches for people to sit and relax. gardens are places with lots of trees and plants.

**Mapping**
*Premise*: Because [Rationale]
*Hypothesis*: The answer of the question "[question]" is [Answer's choice].
*NLI Class*: Entailment

**Mapped example**
*Premise*: Because state park is a [...] gardens are places with lots of trees and plants.
*Hypothesis*: The answer of the question "what is a place that has a bench nestled in trees?" is state park.

Table 2: The mapping design for the four sub-tasks with non-cherry-picked examples. See Section 3 for description about each sub-task.

rationalization model with the training set and generate rationale-answer pairs on the test set. We then gather all predictions from the 60 episodes and randomly select 350 examples for human studies. We present the description of the task, the input instance $x$ and the rationale-answer pair $(\hat{r}, \hat{a})$ for the annotators, and ask them to judge the plausibility of $\hat{r}$, i.e., whether it can justify $\hat{a}$. Following prior works (Marasović et al., 2020; Marasovic et al., 2022), the annotator determines the plausibility by assigning labels from {"*no*", "*weak no*", "*weak yes*", "*yes*"}. We then map labels to plausibility scores {1, 2, 3, 4} and instances with average scores above 2.5 are deemed plausible. We provide inter-annotator agreement details in Appendix C.

The results are shown in Figure 2. We can observe that for all sub-tasks, when the explanations are judged as plausible, they are much more likely paired with correctly predicted answers in constrast to implausible ones. This verifies our hypothesis (discussed in Section 2) and shows plausibility to be a strong signal for correct label predictions. Our results also align with the prior work (Wiegreffe et al., 2021), where they find self-rationalization models demonstrate high label-rationale association against robustness testing. In conclusion, identifying $(\hat{r}, \hat{a})$ pairs that have plausible $\hat{r}$ spurs great potential for boosting model performance, and connects us to *RQ2*.

## 5  Zero-Shot Augmentation of Rationale-Answer Pairs

As shown in Section 4, plausible explanations imply that the corresponding task predictions are more

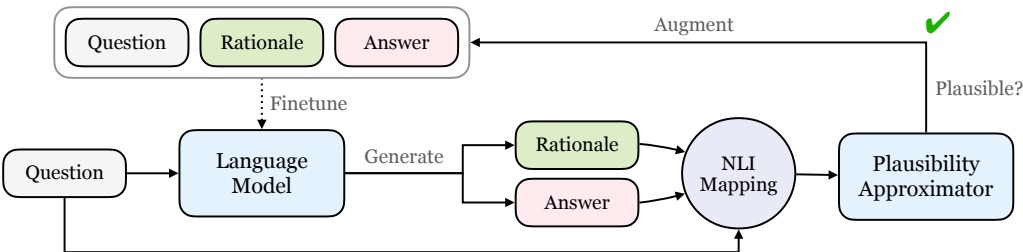

Figure 3: Overview of the self-training paradigm for ZARA. A language model is fine-tuned to generate predictions for unlabeled instances, which are then mapped to NLI formats. The approximator then identifies high-confident (likely plausibile) ones as augmentation for learning an new model.

likely to be correct. Following we present ZARA—the approach towards automatically judging the plausibility of generated explanations, and leverages the high confident rationale-answer pairs to boost model performance via self-training.

## 5.1 Reduce plausibility judgement to NLI

Given a rationale-answer pair $(\hat{r}, \hat{a})$ output by a self-rationalization model, a human evaluates whether $\hat{r}$ is plausible by understanding the input context and the task objective, and applying reasoning ability to determine if $\hat{r}$ justifies $\hat{a}$. Specifically, humans implicitly form propositions from the input context and rationale by understanding the problem (the task). Then do *inference*, i.e., apply logic and reasoning to draw conclusions, in their mind to decide if the propositions support the predicted answer. This mental process of assessing plausibility resembles determining the relationship between a premise and a hypothesis. Driven by this formulation, we reduce the problem of judging the plausibility of explanations to the task of natural language inference (NLI), and construct a zero-shot *approximator*, which leverages existing NLI models to automatically approximate the human judgement of plausibility.

**NLI Mapping.** The formulation as NLI requires the mapping of $(x, \hat{r}, \hat{a}) \rightarrow (p, h)$, where $x$, $p$, and $h$ are the input instance, premise, and hypothesis, respectively. We manually create the mappings for each FEB sub-task as shown in Table 2. Constructing such mappings can be easily achieved with minimal effort [5] compared with human evaluation on all $\hat{r}$. Consider the COMVE example in Table 2, the goal is to select the nonsensical sentence from two sentences. As we can see "*i drove my computer to the gas station.*" is nonsensical, and the rationale

justifies it by stating "*you can't drive a computer.*", which explains why the answer is nonsensical by providing information refuting the answer sentence, resulting in a contradiction relationship between the two. Hence, the approximator can estimate the degree of plausibility by referring to the score of the *contradiction* class.

**The approximator.** For developing the approximator, we ensemble three state-of-the-art pre-trained NLI models by averaging their output scores for the decision of NLI class. Specifically, we adopt RoBERTa (Liu et al., 2019), DeBERTa (He et al., 2020), and BART (Lewis et al., 2020), trained on the MultiNLI corpus (Williams et al., 2018), one of the largest available NLI dataset. The approximator is zero-shot, i.e., all three models are used off-the-shelf (See Appendix A for details) without any fine-tuning on our dataset, accommodating the few-shot, data scarcity setting.

## 5.2 Self-training

In the self-training paradigm, a trained model augments its own training set by constructing pseudo-parallel data with predictions on unlabeled instances, where the most confident predictions are collected as new training examples and used to re-train an improved model. For applying self-training, most works focus on classification tasks (Miyato et al., 2018; Xie et al., 2020; Gera et al., 2022) with common strategies based on operations of confidence scores such as probability values to select new examples. E.g., finding predictions that are far from the decision boundary (Slonim et al., 2011).

However, the adoption of self-training for self-rationalization differs from typical classification tasks in two aspects: (1) Compared with fixed classification labels, the target space of neural sequence generation is much more complex. (2) The selec-

---

[5]One can simply design the mapping by observing the training data.

tion requires considering both the task label $\hat{a}$ and the rationale $\hat{r}$ with their relationship. By a proxy model, i.e, the approximator, we could reduce the target dimensions to fixed class labels to address the former. For the latter, we could resolve it by only considering the plausibility of $\hat{r}$ since plausible $\hat{r}$ likely implies correct $\hat{a}$ as shown in Section 4. Following we introduce our self-training paradigm of ZARA—a *train-judge-train* procedure. See Figure 3 for illustration.

Given an episode $E$ consisting of a training split $\mathcal{D}_{\text{train}}$ and a test split $\mathcal{D}_{\text{test}}$, where an example in $E$ is an input-rationale-answer tuple $(x, r, a)$. We first *train* a LM $M_0$ on $\mathcal{D}_{\text{train}}$ for self-rationalization by prompt-based fine-tuning with natural language prompts. The trained model is denoted as $M_1$.

Next, we perform inference with $M_1$ on unlabeled instances $x \in \mathcal{D}_{\text{unlabled}}$, where $\mathcal{D}_{\text{unlabled}}$ is a non-overlapping set randomly sampled from other episodes with size $|\mathcal{D}_{\text{unlabled}}| = |\mathcal{D}_{\text{test}}|$. For each prediction, the input $x$ and the generated rationale-answer pair $(\hat{r}, \hat{a})$ are mapped to the NLI format, i.e., $(x, \hat{r}, \hat{a}) \rightarrow (p, h)$, and passed to the zero-shot plausibility approximator.[6] The approximator automatically *judges* the plausibility of $\hat{r}$, where the most confident predictions are selected by a plausibility threshold $\alpha$, i.e., a probability score (See Appendix B for details). This process does not require any ground truth label or golden rationale.

The collected high-confident $(x, \hat{r}, \hat{a})$ predictions become new instances to augment $D_{\text{train}}$. Also, we ensure the added instances are balanced for classification tasks by downsampling majority classes. We then re-*train* $M_0$ on the augmented training split to obtain our final self-rationalization model $M_2$, and evaluate on $\mathcal{D}_{\text{test}}$.

# 6 Experiments

In this section, we discuss the experimental setup and present the results of our proposed method, ZARA, for improving few-shot self-rationalization via self-training. We also perform human and quantitative evaluations to validate the automatic plausibility assessment for our approximator.

## 6.1 Model

For comparison purposes, we follow FEB and use UNIFIEDQA (Khashabi et al., 2020), a T5 (Raffel et al., 2020) variant trained on a multi-task

---

[6]Depending on the mapping design, some sub-tasks do not require input content $x$ to form the premise and hypothesis.

mixture of QA datasets, as our self-rationalization model for all experiments. The model performs few-shot learning via fine-tuning with natural language prompts. We experiment with three model sizes: UNIFIEDQA-base (200M), UNIFIEDQA-large (770M), and UNIFIEDQA-3B (2.7B). The results presented in Section 4 are conducted with UNIFIEDQA-3B. More details of the experimental setups and configurations are provided in Appendix A.

| | Method | Model | Acc. | $\Delta$ | BERTsc. | $\Delta$ |
|---|---|---|---|---|---|---|
| **ComVE** | FEB | Base | $67.3_{0.7}$ | 6.8 | $61.0_{0.6}$ | 6.6 |
| | ZARA | Base | $74.1^{\dagger}_{0.5}$ | | $67.6^{\dagger}_{0.5}$ | |
| | FEB | Large | $81.3_{0.4}$ | 5.9 | $73.9_{0.4}$ | 5.8 |
| | ZARA | Large | $87.2^{\dagger}_{0.3}$ | | $79.7^{\dagger}_{0.2}$ | |
| | FEB | 3B | $89.0_{0.4}$ | 4.7 | $81.0_{0.3}$ | 4.9 |
| | ZARA | 3B | $\mathbf{93.7}^{\dagger}_{0.2}$ | | $\mathbf{85.9}^{\dagger}_{0.1}$ | |
| | GPT-3 | INSTRUCTGPT | $74.0_{1.5}$ | – | $67.6_{1.3}$ | – |
| | Random | – | 50.0 | – | – | – |
| **SBIC** | FEB | Base | $67.5_{0.4}$ | 5.4 | $65.3_{0.4}$ | 5.0 |
| | ZARA | Base | $72.9^{\dagger}_{0.2}$ | | $70.3^{\dagger}_{0.2}$ | |
| | FEB | Large | $71.1_{0.4}$ | 3.6 | $68.5_{0.4}$ | 3.5 |
| | ZARA | Large | $74.7^{\dagger}_{0.2}$ | | $72.0^{\dagger}_{0.2}$ | |
| | FEB | 3B | $71.7_{0.5}$ | 4.4 | $68.9_{0.5}$ | 4.4 |
| | ZARA | 3B | $\mathbf{76.1}^{\dagger}_{0.2}$ | | $\mathbf{73.3}^{\dagger}_{0.2}$ | |
| | GPT-3 | INSTRUCTGPT | $74.2_{1.4}$ | – | $71.5_{1.4}$ | – |
| | Random | – | 50.0 | – | – | – |
| **E-SNLI** | FEB | Base | $75.0_{0.3}$ | 3.0 | $67.5_{0.3}$ | 3.0 |
| | ZARA | Base | $78.0^{\dagger}_{0.2}$ | | $70.5^{\dagger}_{0.2}$ | |
| | FEB | Large | $84.8_{0.3}$ | 1.2 | $76.6_{0.3}$ | 1.1 |
| | ZARA | Large | $86.0^{\dagger}_{0.2}$ | | $77.7^{\dagger}_{0.2}$ | |
| | FEB | 3B | $87.4_{0.2}$ | 2.1 | $79.1_{0.2}$ | 2.2 |
| | ZARA | 3B | $\mathbf{89.5}^{\dagger}_{0.2}$ | | $\mathbf{81.3}^{\dagger}_{0.2}$ | |
| | GPT-3 | INSTRUCTGPT | $65.4_{0.5}$ | – | $59.8_{0.5}$ | – |
| | Random | – | 33.3 | – | – | – |
| **ECQA** | FEB | Base | $41.4_{0.3}$ | -1.9 | $36.7_{0.3}$ | -2.0 |
| | ZARA | Base | $39.5_{0.2}$ | | $34.7_{0.2}$ | |
| | FEB | Large | $57.2_{0.4}$ | 0.4 | $51.0_{0.3}$ | 0.0 |
| | ZARA | Large | $57.6_{0.2}$ | | $51.0_{0.2}$ | |
| | FEB | 3B | $65.9_{0.4}$ | 4.1 | $59.0_{0.3}$ | 3.2 |
| | ZARA | 3B | $\mathbf{70.0}^{\dagger}_{0.2}$ | | $\mathbf{62.2}^{\dagger}_{0.2}$ | |
| | GPT-3 | INSTRUCTGPT | $60.6_{1.5}$ | – | $54.4_{1.3}$ | – |
| | Random | – | 20.0 | – | – | – |

Table 3: Experiments comparing FEB (Marasovic et al., 2022) and ZARA, where the results with GPT-3 (175B) and random baseline, i.e., $\frac{1}{number\ of\ classes}$ are also presented. We assess the statistical significance of ZARA's performance gain over the FEB baseline by adopting one-sided McNemar's test (McNemar, 1947), where $\dagger$ denotes $p < 0.01$.

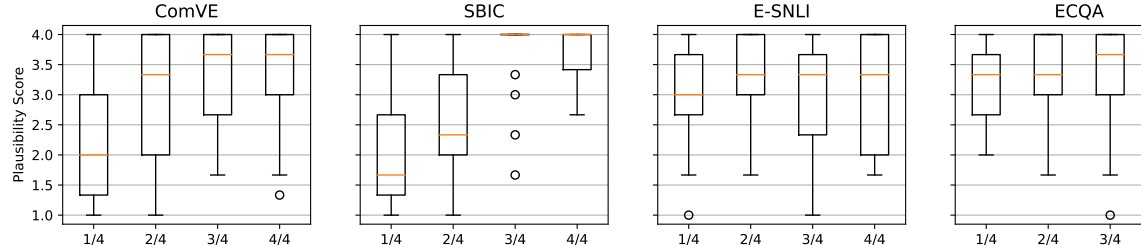

Figure 4: Human judgement of plausibility scores under different pseudo-plausibility percentile bins. $\frac{1}{4}$, $\frac{2}{4}$, $\frac{3}{4}$, and $\frac{4}{4}$ indicate 1~25th, 25~50th, 50~75th, and 75~100th percentile group, respectively. Ranked by the approximator's output probability, i.e., pseudo-plausibility scores.

## 6.2 Main results

The evaluation metrics of FEB are accuracy and BERTscore (Zhang et al., 2019) for end-task labels and explanations, respectively.[7] For each sub-task, we train 60 models (one per episode) and report the mean and standard error of accuracy/BERTscore in Table 3. We also provide statistics on the number of instances added for augmentation in Appendix D. To the best of our knowledge, we present the first results on the newly introduced FEB benchmark (besides their original approach in the paper).

We experiment with three model sizes: base, large and 3B. In ZARA, both training stages adopt models of the same size; the original FEB baseline only involves training one model (one stage). As observed in Table 3, our method substantially outperforms the FEB baseline for all datasets. In general, COMVE, SBIC and E-SNLI demonstrate relatively consistent improvements across model size. The only anomoly is for ECQA. We hypothesize the under-parameterized models (base and large) suffer forgetting from continuous learning with the augmented data (Kirkpatrick et al., 2017), since ECQA may require commonsense knowledge which is not presented in the FEB training data but is encoded in models' parameters originally. However, for the 3B model—which is still significantly smaller than most large-scale LMs—great performance gain with ZARA is exhibited.

## 6.3 Approximator evaluation

**Plausibility evaluation** We conduct human evaluation to validate our approximator. Specifically, the human evaluation can be considered as a meta-evaluation for evaluating the approximator's ability to evaluate explanations, i.e., its ability to assess

plausibility. To recap, the approximator's output probability of the corresponding NLI class (based on the mapping design in Table 2) represents an estimation of plausibility degree, i.e., a pseudo-plausibility score. We use the same batch of annotated data from Section 4. That is, 350 randomly selected examples generated by the stage-one model with human judgement of plausibility value {1, 2, 3, 4} mapped from {"*no*", "*weak no*", "*weak yes*", "*yes*"} and averaged across annotators.

The results are presented in Figure 4. We group the instances into four bins, each containing 25% of data according to the percentile ranking of their pseudo-plausibility score. In general, the median performance of human plausibility judgement increases with higher percentile groups, especially for the COMVE and SBIC sub-tasks. Interestingly, due to the nature of NLI model of the approximator, its output (i.e., pseudo-plausibility scores) may be effected by spurious surface features learned only for NLI tasks (transferred from the MultiNLI dataset), giving rise to the larger interquartile range of the top percentile group in E-SNLI. Overall, the results show our approximator is capable of reflecting human plausibility judgement.

**Correctness evaluation** As stated in Section 4, plausible rationales likely indicate correct answer predictions. We further evaluate our approximator regarding this property by checking the end-task answer accuracy of the data subset selected for augmentation from stage-one model's prediction pool. We consider three selection strategies: (1) ZARA, i.e., our proposed method, which selects confident (high-scoring) predictions; (2) Random, the data subset is selected randomly from prediction pool; (3) Lowest, in contrast to ZARA, we select a subset from the data with lowest-ranking pseudo-plausibility scores.

For each episode, the number of augmented

---

[7]BERTscore is one of the most correlated automatic NLG metrics with human judgement of plausibility for free-text explanation, as shown by Kayser et al. (2021).

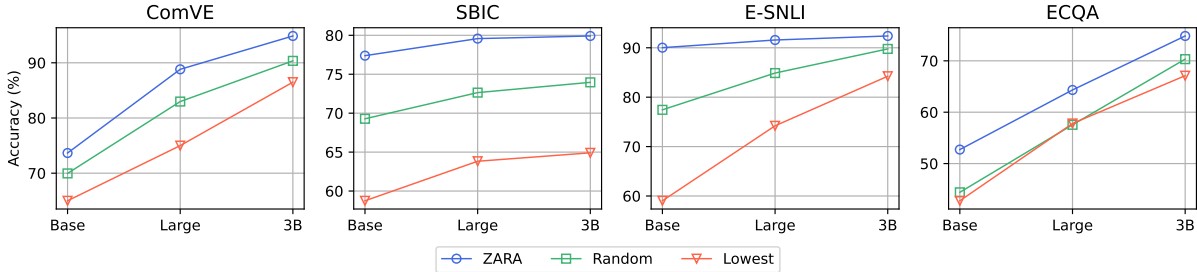

Figure 5: The average accuracy (per episode) of the selected subset from the stage-one model's generated results by different selection strategies: (1) ZARA, (2) Random, and (3) Lowest.

instances for (2) and (3) are determined by (1), i.e., we randomly select $n$ instances or select $n$ bottom-ranking instances, where $n$ is the number of instances for augmentation using ZARA. The results are shown in Figure 5. We can observe ZARA consistently outperforms Random and Lowest with substantial margins under different model sizes across all four datasets, and Lowest demonstrates the poorest accuracy. This suggest our approximator is able to verify label predictions post-hoc, i.e., the high/low pseudo-plausibility score suggests the prediction is accurate/inaccurate. In conclusion, the overall evaluation results suggest our approximator can effectively extract rationale-answer pairs which are more plausible and accurate.

## 7 Related Work

### 7.1 Few-shot self-rationalization

To provide NLEs under low supervision, Marasovic et al. (2022) propose the FEB benchmark and establish the first results by exploring natural language prompt-based fine-tuning. Wiegreffe et al. (2022) focus on improving NLEs with an overgenerate-and-filter pipeline: prompting GPT-3 with gold labels to generate explanation candidates which are then filtered by a model trained with human annotations. Recent works (Wei et al., 2022; Wang et al., 2022b; Huang et al., 2022) leverage rationale-augmented chain-of-thought (CoT) inputs to prompt frozen large-scale LMs in few-shot. Concurrent works (Wang et al., 2022a; Ho et al., 2022; Hsieh et al., 2023) propose pipeline frameworks to distill knowledge by prompting a large "teacher" LM to generate diverse reasoning rationales which are then used to fine-tuning a small "student" LM. In comparison, ZARA directly optimizes small LMs on downstream tasks, without access to any large LMs.

A previous work that shares a conceptual simi-

larity to ours is STaR (Zelikman et al., 2022). Give an initial training set consisting of a large amount of labels and a few seed rationales, STaR iteratively fine-tunes a GPT-J model to build an augmented training set to bootstrap itself. The fundamental difference between ZARA and STaR is that STaR needs ground-truth labels to select and generate rationales for augmentation, whereas, ZARA augments rationale-label pairs in zero-shot, without any requirements of ground-truth labels or golden rationales. Another related work by Ye and Durrett (2022) leverages NLEs to boost end-task predictions post-hoc, via training a calibrator. In comparison, we directly improve self-rationalization and our approximator does not require any further training. Moreover, all LMs used by Ye and Durrett (2022) are 175B.

### 7.2 Leveraging NLI for downstream tasks

The framework of NLI has been expanded to benefit many NLP tasks. Welleck et al. (2019) develop a dataset to improve dialogue models by framing the dialogue consistency problem as NLI. Honovich et al. (2021); Dziri et al. (2022) use NLI to design automatic metrics evaluating factuality of knowledge-grounded dialogue systems. Falke et al. (2019); Kryscinski et al. (2020); Laban et al. (2022) use NLI models to detect factual errors in abstractive summarization tasks. For question answering, Chen et al. (2021) propose a framework to verify QA systems' predictions with NLI by training models to generate premise-hypothesis pairs from QA instances. Driven by human reasoning, Yin et al. (2019) approach text classification in zero-shot by formulating it as an entailment problem—given the input text (premise), humans mentally construct hypotheses "*the text is about* [label choice]" to determine the answer—and adopt out-of-the-box NLI models for predictions.

## 8 Conclusion

In this work, we first show evidences that plausible explanations imply correct label predictions, and leverage a novel NLI approximator to automatically identify plausible rationales paired with correct answers from unlabeled results.s By collecting such rationale-answer pairs with self-training, we effectively improve the performance of few-shot self-rationalization for small LMs. Moreover, we demonstrate the potential for automatic evaluation of free-text explanations. In light of this, we believe developing a supervised approximator with a unified NLI mapping schema across tasks to be a promising avenue for future works.

## Limitations

The success of the approximator relies on the quality of the NLI mapping. Though we showcase great improvement across four different tasks, if the complexity of a task makes the mapping construction non-trivial, the created mapping might not be able to accurately reflect human plausibility judgement of the generated rationales, and the benefit of self-training could not be guaranteed. Namely, the approximator may identify noisy instances that would instead hurt model performance.

## Acknowledgements

We thank the reviewers for their insightful comments. This research was partially supported by National Science and Technology Council, Taiwan, under grants MOST 110-2221-E-002-128-MY3 and NSTC 111-2634-F-002-023-, and Ministry of Education (MOE) in Taiwan, under grants NTU-112L900901.

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

## A   Experimental setups and configurations

We prepare the training data and format it as natural language prompts using the scripts provided by the FEB paper's repository. We train our base and large model (UNIFIEDQA-base and UNIFIEDQA-large) by NVIDIA GeForce RTX 3090, and 3B model (UNIFIEDQA-3B) by NVIDIA RTX A6000. For hyper-parameter (HP) settings, we follow the original setup in the FEB paper for stage-one training, and for stage-two, we set maximum epochs to 48 and keep other HPs the same as stage-one. We do not perform additional HP search.

For the approximator, We use *facebook/bart-large-mnli*, *microsoft/deberta-large-mnli*, and *roberta-large-mnli* from the Hugging Face Hub.

## B   Plausibility threshold

To estimate a threshold that is sufficient but not overly strict, we compute the average number of training set instances (which are required to be plausible) per episode with probability scores above different threshold values, as shown in Figure 6. The dotted line represents the segment with the smallest slope, indicating increasing the threshold results in the largest data lost. The starting, i.e., smaller, $x$-value of the dotted line is chosen as our plausibility threshold. Thus, 0.9 for COMVE, E-SNLI and ECQA, and 0.8 for SBIC.

## C   Human annotation details

We invite three annotators[8] to conduct human evaluation and compute inter-annotator agreements by Randolph's $\kappa$ on 100 overlapping annotation examples. We record $\kappa$ of 0.60, 0.56, 0.36, and 0.49 for COMVE, SBIC, E-SNLI, and ECQA, respectively. The low (0.36) to moderate (0.60, 0.56, 0.49) agreements align with prior works' observations on evaluating plausibility of free-text explanation, reflecting the task subjectivity (Wiegreffe et al.,

---

[8]The annotators include two graduate students and one Ph.D. student. As our tasks do not require specific domain expertise, the payment is determined by the minimum wage.

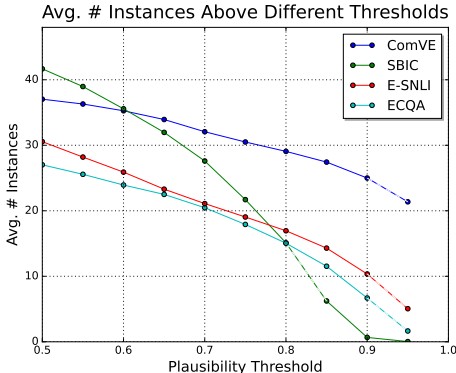

Figure 6: The average number of training set instances (per episode) with probability score above different thresholds.

2022) and could require more fine-grained analysis in the future (Marasovic et al., 2022).

## D   Data augmentation details

Table 4 reports the average number of additional test set instances added per episode for stage-two training. For COMVE, SBIC, and E-SNLI, about one-third of the test data are selected, with only minor differences against model sizes. On the other hand, ECQA shows a notable increment on the 3B model, yet significant lower addition number in general comparing to the other three sub-tasks, which may attribute to the nature of difficulty for commonsense question answering.

| Model | ComVE | SBIC | E-SNLI | ECQA |
|-------|-------|------|--------|------|
| Base  | 109.0 | 113.6 | 123.8 | 13.2 |
| Large | 132.3 | 109.0 | 128.6 | 13.4 |
| 3B    | 191.5 | 107.6 | 98.3  | 48.4 |

Table 4: The comparison for average number of instances added (per episode) between model size.