# OpenReview forum: "ZARA: Improving Few-Shot Self-Rationalization for Small Language Models"
_EMNLP/2023/Conference — EMNLP 2023 Findings_

### Official Review · Reviewer_x5eQ · 2023-08-02

**Soundness:** 4

**Excitement:**

3: Ambivalent: It has merits (e.g., it reports state-of-the-art results, the idea is nice), but there are key weaknesses (e.g., it describes incremental work), and it can significantly benefit from another round of revision. However, I won't object to accepting it if my co-reviewers champion it.

**Paper Topic And Main Contributions:**

In this paper, the authors propose a framework, ZARA, to improve self-rationalization in small language models. The framework consists of an approximator, which converts samples into NLI-type examples and utilizes an ensemble of NLI models to make predictions on unlabeled instances. The samples that are given highly confident scores are used to retrain the language model and improve self-rationalization. The results demonstrate that this framework is able to improve upon existing results in the FEB dataset. In addition, the approximator is able to select unlabeled samples with plausible rationales as produced by the base model, leading to higher accuracy.

**Questions For The Authors:**

Would it not make more sense to use an ensemble of NLI models each trained on different NLI datasets, such as ANLI?

**Reasons To Accept:**

The paper proposes a simple yet effective methodology to find additional instances for self-training language models for improved rationalization.

Rather than focus on improving large language models, the authors focus on improving smaller and more accessible models with their framework.


**Reasons To Reject:**

One concern that I have is that the framework is not completely “automatic”. Instead, mappings must be developed for each subtask that needs to be improved upon.

What is an episode? It says that you train 60 models, one for each episode but I am unclear of the difference between episodes since they all belong to the same subtask in the dataset. Due to the number of models trained, it seems to be rather inefficient in the few-shot setting.


**Reproducibility:**

3: Could reproduce the results with some difficulty. The settings of parameters are underspecified or subjectively determined; the training/evaluation data are not widely available.

**Reviewer Confidence:**

4: Quite sure. I tried to check the important points carefully. It's unlikely, though conceivable, that I missed something that should affect my ratings.

**Typos Grammar Style And Presentation Improvements:**

Line 209: “As described in Section 2, following we attempt”

Section 5.1 has some grammatical errors such as line 259: “a human” and line 463: “humans”

Line 438: “affected”

Line 524: “evidence”

Line 528

---

> ### Author Rebuttal · Authors · 2023-08-29
>
> Thank you for the constructive feedback! Regarding the points mentioned in “Reasons To Reject”:
>
> > One concern that I have is that the framework is not completely “automatic”. Instead, mappings must be developed for each subtask that needs to be improved upon.
>
> - We acknowledge that the mapping design is not fully “automatic”. Though, as discussed in Lines 281-284, constructing such mappings only requires minimal efforts via observing a few instances as opposed to human evaluation of the plausibility of all generated instances. Building an automated framework with zero human-in-the-loop is left as our future work. Thanks for your comment!
>
> > What is an episode? It says that you train 60 models, one for each episode but I am unclear of the difference between episodes since they all belong to the same subtask in the dataset. Due to the number of models trained, it seems to be rather inefficient in the few-shot setting.
>
> - An episode is a train-test split with 48 training examples and 350 evaluation examples, where different episodes are constructed with examples randomly sampled from the full dataset of the subtasks. The goal of such a design is for building a reliable few-shot evaluation setup recommended by “Flex: Unifying evaluation for few-shot nlp (Bragg et al., 2021)” (described in Section 2 of the FEB paper), and in practice, we only need to train one model.
>
> Regarding the “Questions For The Authors”:
>
> > Would it not make more sense to use an ensemble of NLI models each trained on different NLI datasets, such as ANLI?
>
> - Thank you for the suggestion and yes we agree! Ensembling different NLI models trained on different NLI datasets is a promising and insightful setup which we will investigate in the future.
>
> We will fix all typos in the final version, thank you!

---

### Official Review · Reviewer_JA6T · 2023-08-05

**Soundness:** 2

**Excitement:**

3: Ambivalent: It has merits (e.g., it reports state-of-the-art results, the idea is nice), but there are key weaknesses (e.g., it describes incremental work), and it can significantly benefit from another round of revision. However, I won't object to accepting it if my co-reviewers champion it.

**Paper Topic And Main Contributions:**

The paper proposes an approach called ZARA, that is designed to improve self-rationalization capability for smaller models. A filter that separates plausible rationales from generations are added back to the training pool of the model, leading to improvements in the model performance as compared to prior baselines like FEB. This design is motivated by empirical proof that plausibility correlates with correctness of answers.

**Questions For The Authors:**

More critical questions in the Weaknesses section. More questions -
1. L212 Which version of the self-rationalizing model do you use in order to validate your hypothesis in Section 4? (some more details about the exact data split used, model size, number of shots, etc, would be good to add to the paper)
2. Section 5.2: What is the difference between ZARA and STaR (Zelikman et. al) in the self-training approach? How many rationales are actually plausible in STaR vs. the improvement yielded by ZARA?
3. L359 - M_0 is retrained on the augmented training split, why not M_1? It is also confusing if ZARA is continually trained approach? Are there more sequential versions of M?
4. Are improvements in Table 3 significant? (would love to see some significance tests here, specifically for base/large models as they are understudied in self-rationalizing literature)


**Reasons To Accept:**

The paper proposes an excellent yet simple approach to improve accuracy of small self-rationalizing models. The trick to add a rationale filter on the basis of their plausibility is a good design idea that can be easily applied for improved performance. Furthermore, it has been shown in prior work that rationalization emerges as model scales (Wei et. al 2022), so the paper's approach for improving rationalization for smaller models is extremely relevant for the community!




**Reasons To Reject:**

RR1 Motivation: The core idea of the paper lies in L164-168. The paper shows in Section 4 that measuring correlation between plausibility of a rationale correlates with the correctness of the answer. It is unclear if the paper tries to distinguish between *actually plausible* rationales vs. rationales that probably leak labels directly? This comes about as a question specifically because the agreements between annotators is low (which I recognize is similar to prior work). But has the paper looked into *why* plausibility annotations have low agreement?

RR2 Experiments -
1. The standard argument about correlation does not imply causation also holds here. How can one conclusively say that the performance improvements are being shown because of *plausible* rationales being added, rather than simply adding *more* rationales? Clearly, ZARA requires adding more rationales to its training pipeline (almost double than what FEB uses in certain tasks). How does a similar few-shot experiment setting with FEB (with compare number of samples) hold?
2. The way the data splits are created is extremely confusing (L336 onwards). How are D_unlabeled for a given episode taken? If they are randomly sampled from other episodes, doesn't it mean that there can be leaks in the test set on average?
3. Human evaluations are done to validate the 0-shot plausibility approximation. In Figure 4, these judgements are compared. How does the plausibility of M_0 hold with M_1 and M_2? How plausible are FEB's rationales? (It is unclear if there is an actual increase in plausibility of rationales since no baseline plausibility comparisons are made?)

RR3 Human evaluations - Human evaluations are conducted for the plausibility approximator but not for the rationales generated by the final model (M_2)? How does the quality/relevance/association of rationales improve? (given that the eventual purpose of self-rationalization models are to also provide explanations for its predictions).

**Reproducibility:**

3: Could reproduce the results with some difficulty. The settings of parameters are underspecified or subjectively determined; the training/evaluation data are not widely available.

**Reviewer Confidence:**

4: Quite sure. I tried to check the important points carefully. It's unlikely, though conceivable, that I missed something that should affect my ratings.

---

> ### Author Rebuttal · Authors · 2023-08-29
>
> Thank you for the constructive comments! To alleviate your concerns raises in “Reasons To Reject”:
>
> > The core idea of the paper lies in L164-168. The paper shows in Section 4 that measuring correlation between plausibility of a rationale correlates with the correctness of the answer. It is unclear if the paper tries to distinguish between actually plausible rationales vs. rationales that probably leak labels directly? This comes about as a question specifically because the agreements between annotators is low (which I recognize is similar to prior work). But has the paper looked into why plausibility annotations have low agreement?
>
> - We did not conduct fine-grained analysis to distinguish “actually plausible rationales” and “rationales that probably leak labels”, however, our core idea holds since the rationales are not priorly given, where rationales and answers are both generated by the model.
>
> - To the best of our understanding, “rationales that probably leak labels” would not affect the agreements between annotators since if the model indeed generates rationales that leak labels which are contradicting to the predicted answer labels, the annotators should all be able to score these rationales as low plausibility.
>
> - Regarding the reasons why plausibility annotations have low agreement, as stated in Lines 878-880, prior works attribute this to the subjectivity nature of the task and requires further analysis in the future, which we agree. We did not conduct such an analysis as our goal is to show plausible rationales generated by the model have a high probability to be paired with correct labels in the predictions.
> We believe one explanation for low agreement is that different annotators require different levels of *convincement* to agree with the reasons provided by rationales.
>
> > The standard argument about correlation does not imply causation also holds here. How can one conclusively say that the performance improvements are being shown because of plausible rationales being added, rather than simply adding more rationales? Clearly, ZARA requires adding more rationales to its training pipeline (almost double than what FEB uses in certain tasks). How does a similar few-shot experiment setting with FEB (with compare number of samples) hold?
>
> - The goal of this work—self-rationalization—is to generate both rationales and answer labels jointly, thus requiring rationale-label pairs as the training data. We believe there is very little reason to assume that simply adding more data, i.e., rationale-label pairs with potentially wrong labels, could lead to performance improvements, especially in our few-shot, prompt-based fine-tuning setting.
> Also, we validate ZARA’s ability to select rationale-label pairs which have correctly predicted labels as shown in Figure 5 and discussed in Section 6.3.
> And the few-shot setting holds in the sense that the number of used instances with gold rationale-label pairs remains the same as in FEB.
>
> > The way the data splits are created is extremely confusing (L336 onwards). How are D_unlabeled for a given episode taken? If they are randomly sampled from other episodes, doesn't it mean that there can be leaks in the test set on average?
>
> - As described in Lines 343-345, D_unlabeled is non-overlapped with D_test, and is randomly selected from another episode. For better clarity, we will revise our wordings in Line 343-345 and explicitly state that D_unlabeled does not contain instances from D_test. Thank you for pointing this out!
>
> > Human evaluations are done to validate the 0-shot plausibility approximation. In Figure 4, these judgements are compared. How does the plausibility of M_0 hold with M_1 and M_2?
>
> - In our self-training procedure (as elaborate in Section 5.2), the model M_0 refers to the original model before any fine-tuning. After the initial fine-tuning, M_0 becomes M_1, and we adopt M_1 to perform inference on D_unlabeled. Lastly, we again fine-tune M_0 but on the augmented dataset with additional instances selected by ZARA on M_1, and the resulting trained model is M_2. Here the human evaluation present in Figure 4 is to validate the ability of ZARA to select plausible rationales, so the evaluation is performed on rationales generated by M_1.
>
> We address the following two comments jointly as they are relevant and dependent.
>
> > How plausible are FEB's rationales? (It is unclear if there is an actual increase in plausibility of rationales since no baseline plausibility comparisons are made?)
>
> > Human evaluations are conducted for the plausibility approximator but not for the rationales generated by the final model (M_2)? How does the quality/relevance/association of rationales improve? (given that the eventual purpose of self-rationalization models are to also provide explanations for its predictions).
>
> - For a unified comparison of previous work (i.e., FEB) and future work, we assessed the plausibility of rationales by BERTscore and show ZARA outperforms FEB (Table 3). As described in footnote 6, prior studies have shown BERTscore to be one of the most correlated automatic metrics with human judgement of plausibility for rationales (which is also the reason why BERTscore is chosen as FEB’s evaluation metric in the first place). However, we agree an additional human evaluation of the plausibility on M_2's generated rationales would be beneficial, and we plan to incorporate it in our camera-ready version.
>
> Regarding the “Questions For The Authors:”
>
> > Which version of the self-rationalizing model do you use in order to validate your hypothesis in Section 4? (some more details about the exact data split used, model size, number of shots, etc, would be good to add to the paper)
>
> - We validate our hypothesis in Section 4 using model M_1’s generated results.
> For model size, we use the UnifiedQA-3B model (described in Lines 379-380).
> Details including data split used and number of shots are described in Lines 197-199, which is the original FEB benchmark setup.
>
> > What is the difference between ZARA and STaR (Zelikman et. al) in the self-training approach?
>
> - The fundamental difference between ZARA and STaR is that STaR needs ground-truth labels to select and generate rationales for augmentation, whereas, ZARA augments rationale-label pairs without any requirements of ground-truth labels or golden rationales.
>
> > How many rationales are actually plausible in STaR vs. the improvement yielded by ZARA?
>
> - The comparison is not suitable as the goal and setting between STaR and ZARA are different. The objective of STaR is to improve the end-task accuracy given a training set consisting of a large amount of labels and a few seed rationales, where these ground-truth labels are required when training STaR to generate rationales and select augmentation data for self-training. However, the goal of ZARA is to jointly improve rationale quality and end-task accuracy in a few-shot manner, that is, the training data only includes a few rationale-answer pairs. In sum, the STaR framework can not be adopted when only a few rationale-answer pairs are allowed as training data (i.e., in the setting of ZARA).
>
> > M_0 is retrained on the augmented training split, why not M_1? It is also confusing if ZARA is continually trained approach? Are there more sequential versions of M?
>
> - We retrain the initial model M_0 following prior self-training approaches, e.g., STaR (Zelikman et. al). In our setting, ZARA is not (although it can be) a continually trained approach. We achieve great empirical improvements (Table 3) with only one self-training iteration as described in Section 5.2.
>
> > Are improvements in Table 3 significant? (would love to see some significance tests here, specifically for base/large models as they are understudied in self-rationalizing literature)
>
> - Thanks for bringing this up, following we report results of significance tests by adopting one-sided McNemar’s test (McNemar, 1947) to assess the statistical significance of ZARA’s performance gain over the FEB baseline, and “v” denotes p value < 0.01. We will add the details into our final version. As observed, base and large have significant results in 3 out of the 4 subtasks, and 3B is significant in all 4 subtasks.
> - ComVE
>
>     |       | Accuracy | BERTscore |
>     | ----- |:--------:|:---------:|
>     | Base  |    v     |     v     |
>     | Large |    v     |     v     |
>     | 3B    |    v     |     v     |
>
> - SBIC
>
>     |       | Accuracy | BERTscore |
>     | ----- |:--------:|:---------:|
>     | Base  |    v     |     v     |
>     | Large |    v     |     v     |
>     | 3B    |    v     |     v     |
>
> - E-SNLI
>
>     |       | Accuracy | BERTscore |
>     | ----- |:--------:|:---------:|
>     | Base  |    v     |     v     |
>     | Large |    v     |     v     |
>     | 3B    |    v     |     v     |
>
> - ECQA
>     |       | Accuracy | BERTscore |
>     | ----- |:--------:|:---------:|
>     | Base  |         |          |
>     | Large |         |          |
>     | 3B    |    v     |     v     |

---

### Official Review · Reviewer_wxRY · 2023-08-08

**Soundness:** 4

**Excitement:**

4: Strong: This paper deepens the understanding of some phenomenon or lowers the barriers to an existing research direction.

**Paper Topic And Main Contributions:**

This paper presents an approach to few-shot self-rationalization via self-training by demonstrating a clever use of off-the-shelf NLI models for filtering potentially incorrect model predictions and the corresponding rationalization.

**Questions For The Authors:**

- I would like to see more insights on different off-the-shelf NLI models on the the ZARA's eventual performance. Also, it would be interesting to see how the performance compares if based on a more recent NLI model such as WaNLI ("WANLI: Worker and AI Collaboration for Natural Language Inference Dataset Creation", https://wanli.allenai.org/) that is presumably more robust for out-of-domain test cases.

- Why the model size is repeated twice for ZARA in Table 3? Were there different combinations of sizes not included in the paper?

- I found the NLI conversion for ECQA a bit odd due to the insertion of the word "Because" for premise, which makes the sentence incomplete.



**Reasons To Accept:**

- This paper presents a clever use of off-the-shelf NLP models to determine the plausibility of the model predictions combined with the free text rationalization. The proposed approach demonstrates an improvement over previous approaches.

- The paper is well-written and the experiments and analyses are thoughtful.

**Reasons To Reject:**

- No significant reason to reject. The paper presents a solid work that is worthy of acceptance.

**Reproducibility:**

4: Could mostly reproduce the results, but there may be some variation because of sample variance or minor variations in their interpretation of the protocol or method.

**Reviewer Confidence:**

4: Quite sure. I tried to check the important points carefully. It's unlikely, though conceivable, that I missed something that should affect my ratings.

---

> ### Author Rebuttal · Authors · 2023-08-29
>
> Thank you for your positive feedback! Regarding the “Questions For The Authors”:
>
> > I would like to see more insights on different off-the-shelf NLI models on the ZARA's eventual performance. Also, it would be interesting to see how the performance compares if based on a more recent NLI model such as WaNLI ("WANLI: Worker and AI Collaboration for Natural Language Inference Dataset Creation", wanli.allenai.org/) that is presumably more robust for out-of-domain test cases.
>
> - Thank you for the great suggestions, we agree these additional insights would be very interesting and will try to incorporate a discussion on different off-the-shelf NLI models trained with NLI dataset with different characteristics, including the one mentioned (WaNLI).
>
> > Why the model size is repeated twice for ZARA in Table 3? Were there different combinations of sizes not included in the paper?
>
> - In Table 3, we repeat the model size twice to indicate the models used in the two stages (initial stage and self-training stage) of our proposed ZARA, i.e., initial model size - self-training model size. Considering the models used in the two stages are identical, we will simplify the notation in the revised version to reduce confusion.
>
> > I found the NLI conversion for ECQA a bit odd due to the insertion of the word "Because" for premise, which makes the sentence incomplete.
>
> - Thank you for pointing this out! We did not notice this at the time. We agree that the incomplete sentence is a bit odd and could be refined.

---

### Meta-Review · Area_Chair_9Qf8 · 2023-09-16

**Recommendation:** 4

**Metareview:**

This paper proposes a method for self-training of small language models to improve their rationalization capabilities.  They propose to use NLI to evaluate explanation plausibility. Figure 2 establishes that plausible explanations are correlated with correct predictions; if plausible explanations *cause* correct predictions, then training on them to produce more plausible explanations should led to better self-rationalization models. The paper uses this observation to filter model predictions for use in the self-training stage.  Experiments on FEB with a UnifiedQA model show stronger accuracy across the board with the proposed technique.

Reviewers liked the use of off-the-shelf models to improve rationalization. They praised the overall clarity of the paper.

I agree with these strengths and think this is a solid piece of research. However, the differences from STaR don't seem that substantial: the main difference is that ground truth labels are not needed and a different filtering criterion (plausibility) is used. I do believe the authors (in the response to JA6T) that it's hard to properly compare the techniques, so it's more of a conceptual similarity.  However, I would say that this paper should be cited more prominently in the present paper (right now it appears to be "nocited" and only appears in the citations).

JA6T also raises a valid point about causation being derived from data that shows correlation. However, the ultimate test of this is in the final end-to-end results, so I don't see this as a critical soundness weakness. It should be clarified in the paper, though.

---

### Decision · Program_Chairs · 2023-10-07

**Decision:**

Accept-Findings

**Comment:**

This paper proposes a method for self-training of small language models to improve their rationalization capabilities.  They propose to use NLI to evaluate explanation plausibility. Figure 2 establishes that plausible explanations are correlated with correct predictions; if plausible explanations *cause* correct predictions, then training on them to produce more plausible explanations should led to better self-rationalization models. The paper uses this observation to filter model predictions for use in the self-training stage.  Experiments on FEB with a UnifiedQA model show stronger accuracy across the board with the proposed technique.

Reviewers liked the use of off-the-shelf models to improve rationalization. They praised the overall clarity of the paper.

I agree with these strengths and think this is a solid piece of research. However, the differences from STaR don't seem that substantial: the main difference is that ground truth labels are not needed and a different filtering criterion (plausibility) is used. I do believe the authors (in the response to JA6T) that it's hard to properly compare the techniques, so it's more of a conceptual similarity.  However, I would say that this paper should be cited more prominently in the present paper (right now it appears to be "nocited" and only appears in the citations).

JA6T also raises a valid point about causation being derived from data that shows correlation. However, the ultimate test of this is in the final end-to-end results, so I don't see this as a critical soundness weakness. It should be clarified in the paper, though.